# Serious Clinical Outcomes of COVID-19 Related to Acetaminophen or NSAIDs from a Nationwide Population-Based Cohort Study

**DOI:** 10.3390/ijerph20053832

**Published:** 2023-02-21

**Authors:** Jin-Woo Kim, Siyeong Yoon, Jongheon Lee, Soonchul Lee

**Affiliations:** 1Department of Orthopaedic Surgery, Nowon Eulji Medical Center, Eulji University, Seoul 01830, Republic of Korea; 2Department of Orthopaedic Surgery, CHA Bundang Medical Center, School of Medicine, CHA University, Seongnam-si 13488, Republic of Korea

**Keywords:** SARS-CoV-2, COVID-19, acetaminophen, NSAIDs

## Abstract

Acetaminophen and non-steroidal anti-inflammatory drugs (NSAIDs) have been widely prescribed to infected patients; however, the safety of them has not been investigated in patients with serious acute respiratory syndrome coronavirus 2 (SARS-CoV-2) infection. Our objective was to evaluate the association between the previous use of acetaminophen or NSAIDs and the clinical outcomes of SARS-CoV-2 infection. A nationwide population-based cohort study was conducted using the Korean Health Insurance Review and Assessment Database through propensity score matching (PSM). A total of 25,739 patients aged 20 years and older who tested for SARS-CoV-2 were included from 1 January 2015 to 15 May 2020. The primary endpoint was a positive result for a SARS-CoV-2 test, and the secondary endpoint was serious clinical outcomes of SARS-CoV-2 infection, such as conventional oxygen therapy, admission to the intensive care unit, need for invasive ventilation care, or death. Of 1058 patients, after propensity score matching, 176 acetaminophen users and 162 NSAIDs users were diagnosed with coronavirus disease 2019. After PSM, 162 paired data sets were generated, and the clinical outcomes of the acetaminophen group were not significantly different from those of the NSAIDs group. This suggests that acetaminophen and NSAIDs can be used safely to control symptoms in patients suspected of having SARS-CoV-2.

## 1. Introduction

As of December 2019, a new coronavirus, serious acute respiratory syndrome coronavirus 2 (SARS-CoV-2), poses a global health threat. In January 2020, the World Health Organization named the syndrome coronavirus disease 2019 (COVID-19). About 5% of patients with COVID-19 experience acute respiratory distress syndrome (ARDS), septic shock, or multiple organ failure requiring hospitalization in intensive care unit (ICU) [1]. There are several risk factors for mortality from COVID-19, including older age, smoking, cardiovascular disease, chronic kidney disease, diabetes and obesity, malignancy, chronic HIV infection, and treatment with dexamethasone [2,3,4,5,6,7]. Concerns have been raised about drug use related the risk of COVID-19. However, these concerns have not been fully identified.

Acetaminophen (AAP) is a safe analgesic considered as a treatment to reduce fever and chills, which are the first symptoms of COVID-19. AAP and non-steroidal anti-inflammatory drugs (NSAIDs) have been widely prescribed to infected patients to control fever, pain, and inflammation [8]. Both are inexpensive, widely available, and have well-described risk profiles. The main mechanism of NSAIDs is the inhibition of cyclooxygenase enzymes by the formation of prostaglandins derivatives from arachidonic acid [9]. Conversely, NSAIDs treatment for community-acquired pneumonia has been known to be related to an increased risk of pleuropulmonary complications [10]. However, their safety in SARS-CoV-2 patients has not yet been investigated.

Our objective was to evaluate the association between prior use of acetaminophen or NSAIDs and the potential influence on susceptibility to SARS-CoV-2 infection and worsening of serious clinical outcomes of COVID-19 by using nationwide COVID-19 data from the Korean National Health Insurance System (NHIS).

## 2. Materials and Methods

### 2.1. Data Sources and Study Subjects

Data were obtained from the Korean Health Insurance Review and Assessment Service (HIRA). This large-scale cohort provided data on all individuals tested for SARS-CoV-2 in South Korea through services co-operating with the HIRA, the Prevention and Ministry of Health and Welfare, and the Korean Centers for Disease Control (CDC) from 1 January 2015 to 15 May 2020, and referral to the Korean CDC (excluding self-referral) (*n* = 25,739). During the COVID-19 pandemic, the Korean government has provided complementary and compulsory health insurance for all patients with COVID-19. Thus, access to information consisting of personal data, patients’ medical records (including medical visits, prescriptions, diagnoses, and procedures) within 6 years, hospital visits, outcomes related to COVID-19, and death records has been provided in this database of COVID-19. The medical records of all patients were anonymized.

### 2.2. Study Population

We defined the first SARS-CoV-2 test data set for each patient as the cohort entry date (individual index date). Of the 25,739 patients tested for SARS-CoV-2, those under the age of 20, with no history of AAP or NSAIDs treatment, or with a history of prescribing AAP and NSAIDs within 2 weeks of the index date were excluded (*n* = 24,508).

The SARS-CoV-2 infection was defined as a positive real-time reverse transcriptase-PCR (RT-PCR) assay using nasal and pharyngeal swabs according to the World Health Organization (WHO) guidelines [11]. Between 1 January 2015 and 15 May 2020, information on age, sex, and region of residence was extracted from the insurance eligibility data by combining the claims-based data of the National Health Insurance Service. A history of underlying diseases (hypertension—HTN, chronic kidney disease—CKD, cerebrovascular disease—CVA, diabetes mellitus—DM, chronic obstructive pulmonary diseases—COPD, and asthma) was confirmed by submitting at least two claims within one year using the appropriate International Classification of Diseases, 10th revision (ICD-10) code [12]. The Charlson Comorbidity Index (CCI) scores were calculated from the ICD-10 codes using previous methods [12]. The residential region was classified as Seoul, Gyeonggi, Gyeongbuk, Daegu, or other [13]. Drugs used within 30 days prior to the index date included systemic steroids [14].

The final sample included patients who tested for SARS-CoV-2 and were prescribed AAP or NSAIDs, and comprised 1231 individuals, of whom 338 tested positive for SARS-CoV-2 (Figure 1).

### 2.3. Exposure

All prescription AAP or NSAIDs were identified within two weeks of the index date. A non-treatment user was defined as a patient who had not been prescribed AAP or NSAIDs within 2 weeks prior to the index date.

### 2.4. Outcomes

The primary outcome was defined as a positive result for SARS-CoV-2 test [15]. The secondary outcomes were serious clinical outcomes, including composite endpoint 1 (conventional oxygen therapy, admission to the intensive care unit—ICU, mechanical ventilation, or death). In addition, except conventional oxygen therapy, composite endpoint 2 (ICU admission, mechanical ventilation, or death) was analyzed [3].

We analyzed the period from taking the study medication to clinical outcome in patients with COVID-19.

### 2.5. Ethics Approval

This study was approved by the Institutional Review Board of the corresponding author’s hospital. The anonymized data were provided to the authors by NHIS.

### 2.6. Statistical Analysis

A logistic regression model was used to adjust for age, gender, and region of residence (Seoul, Gyeonggi, Gyeongbuk, Daegu, or other) by performing two rounds of propensity score matching (PSM) to balance the baseline characteristics of both groups and decline potential confounding factors; history of HTN, CKD, CVA, DM, COPD, or asthma; CCI (0, 1, or ≥2); and current systemic steroids use. We evaluated the PSM of both groups in a 1:1 ratio using a ‘greedy nearest-neighbor’ algorithm and calculated the predicted probability of AAP versus NSAIDs in all patients for SARS-CoV-2 test (*n* = 25,739) and AAP versus NSAIDs users among patients with confirmed COVID-19 (*n* = 338). Matching adequacy in the absence of major imbalances for each baseline covariate was assessed by comparing the standardized mean difference (SMD) with the distribution of PSM scores, which was more useful than calculating the *p*-values of the t-tests [12]. The primary endpoint was positive results of the SARS-CoV-2 test. The secondary endpoint was the composite endpoint and serious clinical outcomes of COVID-19 patients. Data were analyzed using a logistic regression model and expressed as adjusted ORs (aOR) with 95% confidence intervals (CI) for both groups after adjusting for potential confounding factors; age, sex, region of residence, history of HTN, CKD, CVA, DM, COPD, or asthma; CCI, and current use of systemic steroids. Further analyses were conducted to establish the robustness of the results; AAP or NSAIDs use was stratified by duration of use.

## 3. Results

Of the 25,739 patients who underwent SARS-CoV-2 tests, 1231 patients prescribed either AAP (*n* = 643) or NSAIDs (*n* = 588) were defined in the complete unmatched cohort. In addition, the baseline characteristics in the entire cohort are shown in Table 1. The mean age of the entire cohort was 55.8 years (±19.7 years), and there were 681 females (55.3%).

In the two cohorts, patients taking AAP or NSAIDs were matched in equal numbers (*n* = 529). There are no major imbalances in demographic and no clinical characteristics were observed in SMD within groups of the PSM-matched cohorts. The SARS-CoV-2 test positivity rate in patients using AAP was 33.3% (176/529) compared to 31.0% (162/529) in those using NSAIDs (Table 2).

Table 3 shows the baseline characteristics of COVID-19 confirmed patients.

We performed a PSM-matched analysis of positive SARS-CoV-2 patients. COVID-19 patients had a concordant history of AAP (*n* = 176) and NSAIDs (*n* = 162) use. The baseline characteristics of patients with a diagnosis of COVID-19 treated with AAP or NSAIDs are described in Table 3.

No major imbalances in demographics and clinical characteristics were noted when evaluated using SMD within the PSM-matched cohort groups in Table 4.

The use of AAP was not related to an increased risk of composite endpoint 1 of COVID-19 compared to the use of NSAIDs (Table 5a).

The use of AAP and NSAIDs was not significantly associated with an increased risk of serious COVID-19 outcomes (composite endpoint 1) (Table 5b).

As shown in Table 6, there were no significant differences in the period from taking the medication to clinical outcomes between the AAP and NSAIDs groups.

## 4. Discussion

The present study using a nationwide Korean cohort investigated whether AAP or NSAIDs use increased susceptibility to SARS-CoV-2 infection among 25,739 patients who tested for SARS-CoV-2. This study found that 338 of 1058 patients previously prescribed AAP or NSAIDs had a positive test for SARS-CoV-2. In addition, the study found no significant differences in mortality or serious clinical outcomes in patients receiving AAP or NSAIDs within 2 weeks prior to diagnosis of COVID-19. Our results suggest that the use of AAP or NSAIDs may be a safe option for symptom relief, even when COVID-19 is suspected.

The effect of NSAIDs in patients with COVID-19 has been controversial in previous studies. Prada et al. have demonstrated that exposure to NSAIDs does not increase the risk of SARS-CoV-2 infection or the severity of the COVID-19 [16]. A prospective, multicenter cohort study in the United Kingdom based on the ISARIC Clinical Characterization Protocol [17] demonstrated that NSAIDs use was not associated with worse in-hospital mortality (matched OR 0.95, 95% CI 0.84–1.07; *p* = 0.35), critical care admission (1.01, 0.87–1.17; *p* = 0.89), requirement for invasive ventilation (0.96, 0.80–1.17; *p* = 0.69), or oxygen requirement (1.00, 0.89–1.12; *p* = 0.97). In addition, in a recent systematic review, Zhao et al. [18] also demonstrated that prior use of NSAIDs was not associated with mechanical ventilation, but with a decrease in mortality (aOR), 0.68; 95% confidence interval (CI), 0.52–0.89). Huh et al. revealed that NSAIDs were not related to the diagnosis of COVID-19 (adjusted OR—aOR, 1.04; 95% confidence interval—CI, 0.97–1.12), but were associated with severe disease (aOR, 1.53; 95% CI, 1.25–1.86) using the Korean HIRA database [19].

There have been several studies that revealed the efficacy of different types of NSAIDs. In a double-blinded randomized control study, 500 mg of naproxen every 12 h could improve cough and shortness of breath in COVID-19 patients [20]. In an in vitro study [21], compared to paracetamol or the COX-2 inhibitor celecoxib, naproxen has direct antiviral activity against SARS-CoV-2 replication and protects the lung epithelium from damage caused by the pandemic virus, combining antiviral and anti-inflammatory effects. Another study reported the effectiveness of an in vitro study according to the dose of indomethacin; the treatment with sustained-release formulation at a dose of 75 mg twice daily is expected to achieve a complete response within 3 days for the SARS-CoV-2 infection. [22] Moreover, Kiani et al. [23] investigated the effectiveness of ketotifen, naproxen, and indomethacin, alone or in combination, in reducing SARS-CoV-2 replication. They found that the combination of ketotifen with indomethacin or naproxen all increased in percentage the inhibition of SARS-CoV-2 replication, and no cytotoxic effects were observed. Although this study did not analyze whether different types of NSAIDs affect serious clinical outcomes, it was found that NSAIDs had a positive effect on COVID-19 infection when reviewing previous studies.

Acetaminophen, compared to other over-the-counter drugs, is a safe and commonly recommended analgesic. Micallef et al. [24] demonstrated that symptomatic treatment with NSAIDs for uncomplicated symptoms (fever, pain, or myalgia) deriving from COVID-19 is not recommended due to an increased risk of severe bacterial complications, and treatment with AAP as a safer drug alternative is recommended.

Although not discussed in this study, another study revealed that patients with acute liver injury usually have undetectable levels of AAP; thus, acute liver injury or failure should be considered in patients with COVID-19 when chronic AAP ingestion is reported and is very high [25].

This study using the Korean HIRA database demonstrated that NSAIDs compared to AAP could be an alternative option for the relief of COVID-19 symptoms. NSAIDs can lead to misdiagnosis by masking the fever and worsening the prognosis of COVID-19. It is also possible that ibuprofen may upregulate angiotensin-converting enzyme-2 (ACE-2) expression and allow the SARS-CoV-2 virus to enter easily through epithelial cells.

This study has several advantages. Above all, the data source of the HIRA database consisted of a very large sample data set, and the effects of confounding factors associated with NSAIDs were ruled out using PSM. Furthermore, this was the first study to evaluate the effectiveness of AAP and NSAIDs in Asian COVID-19 patients using PSM and a unique analysis.

There were several limitations in this study. First, patients were included according to prescription medications; therefore, the use of a drug listed on electronic health records may not demonstrate exhaustive exposure to drugs. Second, this was a retrospective study, and, despite efforts to adjust for all confounders by PSM, additional unmeasured confounding factors might have influenced the outcomes. Third, in this study, the total amount of AAP or NSAIDs and different types of NSAIDs were not considered. Despite these limitations, this study revealed evidence based on cohort data of the safety of AAP or NSAIDs prior to the diagnosis of COVID-19.

## 5. Conclusions

The use of AAP or NSAIDs prior to the diagnosis of COVID-19 was not associated with worse outcomes of COVID-19 in a nationwide Korean cohort study with a PSM. Therefore, AAP or NSAIDs can be safely prescribed to COVID-19 patients.

## Figures and Tables

**Figure 1 ijerph-20-03832-f001:**
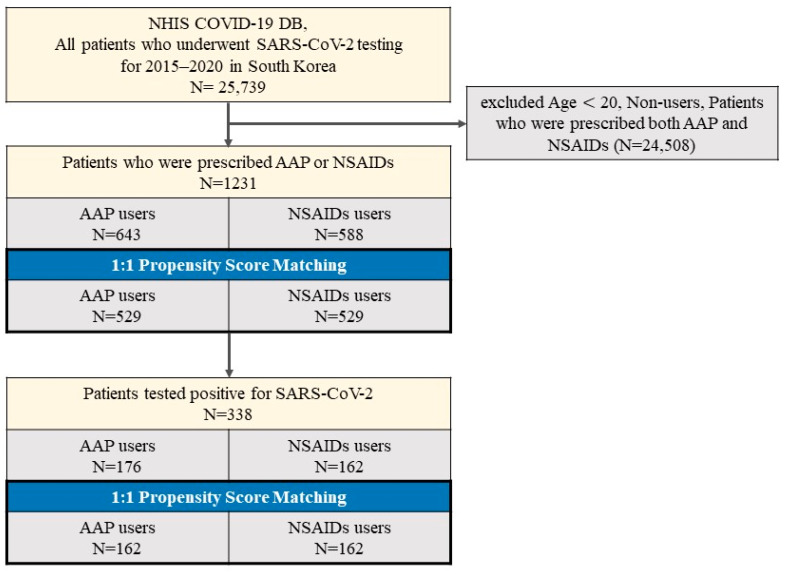
Disposition of patients in the Korean Nationwide Cohort.

**Table 1 ijerph-20-03832-t001:** Baseline characteristics of patients who were prescribed AAP or NSAIDs in the NHIS COVID-19 DB.

Characteristic	Entire Cohort *n* = 1231	AAP *n* = 643	NSAIDs *n* = 588
Sex, *n* (%)			
male	550 (44.7)	286 (44.5)	264 (44.9)
female	681 (55.3)	357 (55.5)	324 (55.1)
Age, *n* (%)			
20–29	149 (12.1)	67 (10.4)	82 (13.9)
30–39	152 (12.3)	66 (10.3)	86 (14.6)
40–49	166 (13.5)	82 (12.8)	84 (14.3)
50–59	227 (18.4)	111 (17.3)	116 (19.7)
60–69	205 (16.7)	109 (17.0)	96 (16.3)
70–79	197 (16.0)	126 (19.6)	71 (12.1)
80–	135 (11.0)	82 (12.8)	53 (9.0)
Region, *n* (%)			
Seoul	196 (15.9)	113 (17.6)	83 (14.1)
Gyeonggi	168 (13.6)	94 (14.6)	74 (12.6)
Daegu	334 (27.1)	179 (27.8)	155 (26.4)
Gyeongbuk	128 (10.4)	71 (11.0)	57 (9.7)
Others	405 (32.9)	186 (28.9)	219 (37.2)
HTN, *n* (%)	458 (37.2)	284 (44.2)	174 (29.6)
COPD, *n* (%)	72 (5.8)	35 (5.4)	37 (6.3)
Asthma, *n* (%)	203 (16.5)	93 (14.5)	110 (18.7)
CKD, *n* (%)	111 (9.0)	82 (12.8)	29 (4.9)
DM, *n* (%)	326 (26.5)	196 (30.5)	130 (22.1)
CVA, *n* (%)	170 (13.8)	107 (16.6)	63 (10.7)
Charlson Comorbidity Index, *n* (%)			
0	413 (33.5)	191 (29.7)	222 (37.8)
1	216 (17.5)	123 (19.1)	93 (15.8)
2 or more	602 (48.9)	329 (51.2)	273 (46.4)
Current use of medication, *n* (%)			
Steroid	254 (20.6)	136 (21.2)	118 (20.1)

AAP: acetaminophen; NSAIDs: non-steroidal anti-inflammatory drugs; HTN: hypertension; COPD: chronic obstructive pulmonary disease; CKD: chronic kidney disease; DM: diabetes mellitus; CVA: cerebrovascular disease.

**Table 2 ijerph-20-03832-t002:** Propensity score-matched baseline characteristics and SARS-CoV-2 infection test positivity between AAP and NSAIDs groups.

Characteristic	AAP *n* = 529	NSAIDs *n* = 529	SMD
Sex, *n* (%)			0.027
male	235 (44.4)	242 (45.7)	
female	294 (55.6)	287 (54.3)	
Age, *n* (%)			0.096
20–29	67 (12.7)	75 (14.2)	
30–39	64 (12.1)	83 (15.7)	
40–49	78 (14.7)	84 (15.9)	
50–59	101 (19.1)	76 (14.4)	
60–69	84 (15.9)	94 (17.8)	
70–79	84 (15.9)	68 (12.9)	
80–	51 (9.6)	49 (9.3)	
Region, *n* (%)			0.091
Seoul	81 (15.3)	71 (13.4)	
Gyeonggi	143 (27.0)	146 (27.6)	
Daegu	77 (14.6)	64 (12.1)	
Gyeongbuk	60 (11.3)	55 (10.4)	
Others	168 (31.8)	193 (36.5)	
HTN, *n* (%)	184 (34.8)	167 (31.6)	0.069
COPD, *n* (%)	29 (5.5)	34 (6.4)	0.039
Asthma, *n* (%)	79 (14.9)	94 (17.8)	0.074
CKD, *n* (%)	34 (6.4)	28 (5.3)	0.050
DM, *n* (%)	126 (23.8)	119 (22.5)	0.032
CVA, *n* (%)	60 (11.3)	60 (11.3)	0.000
Charlson Comorbidity Index, *n* (%)			0.058
0	184 (34.8)	201 (38.0)	
1	90 (17.0)	84 (15.9)	
2 or more	255 (48.2)	244 (46.1)	
Current use of medication, *n* (%)			
Steroid	102 (19.3)	105 (19.8)	0.014

AAP: acetaminophen; NSAIDs: non-steroidal anti-inflammatory drugs; HTN: hypertension; COPD: chronic obstructive pulmonary disease; CKD: chronic kidney disease; DM: diabetes mellitus; CVA: cerebrovascular disease.

**Table 3 ijerph-20-03832-t003:** Baseline characteristics of patients with confirmed laboratory COVID-19 in the NHIS COVID-19 DB.

Characteristic	Entire Cohort *n* = 338	AAP *n* = 176	NSAIDs *n* = 162
Sex, *n* (%)			
male	140 (41.4)	80 (45.5)	60 (37.0)
female	198 (58.6)	96 (54.5)	102 (63.0)
Age, *n* (%)			
20–29	46 (13.6)	22 (12.5)	24 (14.8)
30–39	31 (9.2)	18 (10.5)	13 (8.0)
40–49	50 (14.8)	27 (15.3)	23 (14.2)
50–59	81 (24.0)	43 (24.4)	38 (23.5)
60–69	66 (19.5)	27 (15.3)	39 (24.1)
70–79	38 (11.2)	26 (14.8)	12 (7.4)
80–	26 (7.7)	13 (7.4)	13 (8.0)
Region, *n* (%)			
Seoul	26 (7.7)	13 (7.4)	13 (8.0)
Gyeonggi	22 (6.5)	12 (6.8)	10 (6.2)
Daegu	184 (54.4)	96 (54.5)	88 (54.3)
Gyeongbuk	53 (15.7)	29 (16.5)	24 (14.8)
Others	53 (15.7)	26 (14.8)	27 (16.7)
HTN, *n* (%)	88 (26.0)	50 (28.4)	38 (23.5)
COPD, *n* (%)	6 (1.8)	2 (1.1)	4 (2.5)
Asthma, *n* (%)	47 (13.9)	21 (11.9)	26 (16.0)
CKD, *n* (%)	12 (3.6)	6 (3.4)	6 (3.7)
DM, *n* (%)	69 (20.4)	38 (21.6)	31 (19.1)
CVA, *n* (%)	24 (7.1)	12 (6.8)	12 (7.4)
Charlson Comorbidity Index, *n* (%)			
0	150 (44.4)	74 (42.0)	76 (46.9)
1	51 (15.1)	26 (14.8)	25 (15.4)
2 or more	137 (40.5)	76 (43.2)	61 (37.7)
Current use of medication, *n* (%)			
Steroid	39 (11.5)	21 (11.9)	18 (11.1)

AAP: acetaminophen; NSAIDs: non-steroidal anti-inflammatory drugs; HTN: hypertension; COPD: chronic obstructive pulmonary disease; CKD: chronic kidney disease; DM: diabetes mellitus; CVA: cerebrovascular disease.

**Table 4 ijerph-20-03832-t004:** Propensity score-matched baseline characteristics and clinical outcomes of COVID-19 between AAP and NSAIDs groups.

Characteristic	AAP *n* = 162	NSAIDs *n* = 162	SMD
Sex, *n* (%)			0.064
male	65 (40.1)	60 (37.0)	
female	97 (59.9)	102 (63.0)	
Age, *n* (%)			0.028
20–29	21 (13.0)	24 (14.8)	
30–39	17 (10.5)	13 (8.0)	
40–49	25 (15.4)	23 (14.2)	
50–59	39 (24.1)	38 (23.5)	
60–69	24 (14.8)	39 (24.1)	
70–79	21 (13.0)	12 (7.4)	
80–	15 (9.3)	13 (8.0)	
Region, *n* (%)			0.046
Seoul	12 (7.4)	13 (8.0)	
Gyeonggi	89 (54.9)	10 (6.2)	
Daegu	12 (7.4)	88 (54.3)	
Gyeongbuk	26 (16.0)	24 (14.8)	
Others	23 (14.2)	27 (16.7)	
HTN, *n* (%)	44 (27.2)	38 (23.5)	0.087
COPD, *n* (%)	3 (1.9)	4 (2.5)	0.040
Asthma, *n* (%)	22 (13.6)	26 (16.0)	0.067
CKD, *n* (%)	9 (5.6)	6 (3.7)	0.098
DM, *n* (%)	30 (18.5)	31 (19.1)	0.016
CVA, *n* (%)	16 (9.9)	12 (7.4)	0.094
Charlson Comorbidity Index, *n* (%)			0.020
0	75 (46.3)	76 (46.9)	
1	24 (14.8)	25 (15.4)	
2 or more	63 (38.9)	61 (37.7)	
Current use of medication, *n* (%)			
Steroid	19 (11.7)	18 (11.1)	0.020

AAP: acetaminophen; NSAIDs: non-steroidal anti-inflammatory drugs; HTN: hypertension; COPD: chronic obstructive pulmonary disease; CKD: chronic kidney disease; DM: diabetes mellitus; CVA: cerebrovascular disease.

**Table 5 ijerph-20-03832-t005:** Clinical outcomes of COVID-19 between AAP and NSAIDs.

Variables	AAP *n* = 162	NSAIDs *n* = 162	*p*-Value *
**(a) Risk of composite endpoint 1 and 2**			
Composite endpoint 1, *n* (%)			0.776
No	130 (80.2)	133 (82.1)	
Yes	32 (19.8)	29 (17.9)	
Composite endpoint 2, *n* (%)			0.587
No	143 (88.3)	147 (90.7)	
Yes	19 (11.7)	15 (9.3)	
**(b) Risk or serious COVID-19 outcomes (composite endpoint 1)**			
Conventional oxygen therapy			0.874
No	138 (85.2)	140 (86.4)	
Yes	24 (14.8)	22 (13.6)	
Intensive care unit			0.720
No	159 (98.1)	157 (96.9)	
Yes	3 (1.9)	5 (3.1)	
Mechanical ventilation			1.000
No	154 (95.1)	155 (95.7)	
Yes	8 (4.9)	7 (4.3)	
Death			0.344
No	150 (92.6)	155 (95.7)	
Yes	12 (7.4)	7 (4.3)	

*: Pearson’s chi-square test; composite endpoint 1: conventional oxygen therapy, intensive care unit, mechanical ventilation, or death.; composite endpoint 2: intensive care unit, mechanical ventilation, or death.

**Table 6 ijerph-20-03832-t006:** Period from taking medication to clinical outcome in patients with confirmed laboratory COVID-19.

Clinical Outcome	AAP/NSAIDs *n* (%)	Period Median Days (IQR)		*p*-Value
Conventional oxygen therapy	24 (14.8)/22 (13.6)	11.5 (5.0–16.5)	9.0 (6.0–14.0)	0.628
Intensive care unit	3 (1.9)/5 (3.1)	22.0 (13.0–40.5)	11.0 (10.0–35.0)	1.000
Mechanical ventilation	8 (4.9)/7 (4.3)	6.0 (4.0–20.0)	8.0 (7.0–9.0)	0.521
Death	12 (7.4)/7 (4.3)	25.5 (10.1–31.5)	15.0 (14.5–28.5)	1.000

AAP: acetaminophen; NSAIDs: non-steroidal anti-inflammatory drugs.

## Data Availability

The data that support the findings of this study are included within the article.

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
