# Peer review of "Serious Clinical Outcomes of COVID-19 Related to Acetaminophen or NSAIDs from a Nationwide Population-Based Cohort Study"

_ijerph, 2023, doi:10.3390/ijerph20053832_

Round 1

Reviewer 1 Report

This manuscript focused on the analysis the association of acetaminophen or non-steroidal anti-inflammatory drugs (NSAIDs) with COVID-19 related outcomes from a nationwide population-based cohort using the Korean Health Insurance Review and Assessment database. Although this study has some limitations such as the dose regimen of treatment with acetaminophen or NSAIDs which would be interesting to consider since their properties as anti-inflammatory drugs are different. The authors found that acetaminophen or NSAIDs can be used safely to control symptoms in SARS-CoV-2 patients. However, the manuscript contains some concerns that need to be addressed before considering publication.

Major concerns

1) Materials and methods. Data sources and study subjects. The period included in the abstract (January 1, 2015, to May 15, 2020) is different from the one mentioned in the materials and methods section (January 1, 2015, to December 31, 2020). Please clarify.

2) Materials and methods. Data sources and study subjects. Line 63. The value of n=25,379 is different from that mentioned in the rest of the manuscript (n=25,739). The authors mention that self-referrals were excluded, is this the reason for the difference or is it an error?

3) Materials and methods. Study population. Line 73. The excluded data (n=21,419) do not correspond to those indicated in Figure 1. There is also an error in the age.

4) Materials and methods. Outcomes. The secondary outcomes include composite endpoint 1 and endpoint 2 with the same criteria, it is confusing. The description in the manuscript should be indicated.

5) Materials and methods. Statistical analysis. Line 114. The value of n=25,376. Could you explain about this difference?

6) Results. Line 128. The value of n=558 is different from that mentioned in the table 1 (n=588). The authors should perform a detailed review of the values in table 1. Some errors were detected, for example in the values of the entire cohort, 227(13.5), 197 (16.7), 326 (13.8)

7) Results. Line 139. The authors should perform a detailed review of the percentage values.

8) Results. The authors should perform a detailed review of the values in table 3. Some errors were detected, for example in the values of the NSAIDs, 102(83).

9) Results. The authors should perform a detailed review of the values in table 4. Some errors were detected, for example the values for n=529 do not correspond. Revise the value 102(83.0).

10) Results. Table 5-1. The authors should perform a detailed review of the values in composite endpoint 1.

11) Results. The authors should perform a detailed review of the values of the table 5-2 in mechanical ventilation because are different in table 6.

12) Discussion. In this section you should be improved based on the information that has been written in other parts of the manuscript. Some points should be revised since the authors mention that the worse outcomes were found in 1,231 patients with COVID-19, however the number of patients in this category is different. They also mention that they evaluated the effectiveness of acetaminophen and NSAIDs, but only evaluated some points related with safety.

13) Check once again the whole manuscript for any linguistic problems and correct typo mistakes. For example, there are some errors in the spelling of the abbreviation NSAIDs.

Author Response

This manuscript focused on the analysis the association of acetaminophen or non-steroidal anti-inflammatory drugs (NSAIDs) with COVID-19 related outcomes from a nationwide population-based cohort using the Korean Health Insurance Review and Assessment database. Although this study has some limitations such as the dose regimen of treatment with acetaminophen or NSAIDs which would be interesting to consider since their properties as anti-inflammatory drugs are different. The authors found that acetaminophen or NSAIDs can be used safely to control symptoms in SARS-CoV-2 patients. However, the manuscript contains some concerns that need to be addressed before considering publication.

→ Thank you for your valuable comments.

1) Materials and methods. Data sources and study subjects. The period included in the abstract (January 1, 2015, to May 15, 2020) is different from the one mentioned in the materials and methods section (January 1, 2015, to December 31, 2020). Please clarify.

→ We have corrected ‘December 31, 2020’ to ‘May 15’ in line 62.

2) Materials and methods. Data sources and study subjects. Line 63. The value of n=25,379 is different from that mentioned in the rest of the manuscript (n=25,739). The authors mention that self-referrals were excluded, is this the reason for the difference or is it an error?

→ We have corrected ‘n=25,379’ to ‘n=25,739’ in line 63.

3) Materials and methods. Study population. Line 73. The excluded data (n=21,419) do not correspond to those indicated in Figure 1. There is also an error in the age.

→ We have corrected number of excluded data and an error in the age in Table 1.

4) Materials and methods. Outcomes. The secondary outcomes include composite endpoint 1 and endpoint 2 with the same criteria, it is confusing. The description in the manuscript should be indicated.

→ Composite endpoint 1 included conventional oxygen therapy, admission to the ICU, mechanical ventilation, or death. And composite endpoint 2 included admission to the ICU, mechanical ventilation, or death except conventional oxygen therapy. As you mentioned, it is confusing, so we separately explained and revised composite endpoint 1 and 2 each.

‘The secondary outcomes was serious clinical outcomes included composite endpoint 1 (conventional oxygen therapy, admission to the intensive care unit [ICU], mechanical ventilation, or death) and composite endpoint 2 (ICU admission, mechanical ventila-tion, or death).’ to ‘The secondary outcomes were serious clinical outcomes included composite endpoint 1 (conventional oxygen therapy, admission to the intensive care unit [ICU], mechanical ventilation, or death). And except conventional oxygen therapy, composite endpoint 2 (ICU admission, mechanical ventilation, or death) was analyzed.’

5) Materials and methods. Statistical analysis. Line 114. The value of n=25,376. Could you explain about this difference?

→ We have corrected ‘n=25,376’ to ‘n=25,379’ in line 114.

6) Results. Line 128. The value of n=558 is different from that mentioned in the table 1 (n=588). The authors should perform a detailed review of the values in table 1. Some errors were detected, for example in the values of the entire cohort, 227(13.5), 197 (16.7), 326 (13.8)

→ We have corrected ‘n=558’ to ‘n=588’ in line 128. We have checked and corrected the values in Table 1, ‘n=227(13.5)’ to ‘n=227(18.4)’, ‘n=197(16.7)’ to ‘n=197(16.0)’, and ‘n=326(13.8)’ to ‘n=362(26.5)’.

7) Results. Line 139. The authors should perform a detailed review of the percentage values.

→ We have corrected that ‘The SARS-CoV-2 test positivity rate in patients with AAP was 3.3% (176/529) compared to 3.1% (162/529) in those with NSAIDs.’ to ‘The SARS-CoV-2 test positivity rate in patients with AAP was 33.3% (176/529) compared to 31.0% (162/529) in those with NSAIDs.’

8) Results. The authors should perform a detailed review of the values in table 3. Some errors were detected, for example in the values of the NSAIDs, 102(83).

→ We have checked and corrected the values in Table 3, ‘n=102(83.0)’ to ‘n=102(63.0)’

9) Results. The authors should perform a detailed review of the values in table 4. Some errors were detected, for example the values for n=529 do not correspond. Revise the value 102(83.0).

→ In Table 4, the entire cohort of AAP and NSAIDs were 162 each. We have checked and corrected the values in Table 3, ‘n=102(83.0)’ to ‘n=102(19.3)’.

10) Results. Table 5-1. The authors should perform a detailed review of the values in composite endpoint 1.

→ We have corrected the values in composite endpoint 1, ‘n=130(80.2)’ to ‘n=130(80.0)’ and ‘n=12(80.0)’ to ‘n=32(20.0)’, and ‘n=56(45.2)’ to ‘n=29(17.9)’

11) Results. The authors should perform a detailed review of the values of the table 5-2 in mechanical ventilation because are different in table 6.

→ As you mentioned, we have corrected the values of the table 5-2 in mechanical ventilation, ‘n=3(1.9)’ to ‘n=8(4.9)’ and ‘n=5(3.1)’ to ‘n=7(4.3)’.

12) Discussion. In this section you should be improved based on the information that has been written in other parts of the manuscript. Some points should be revised since the authors mention that the worse outcomes were found in 1,231 patients with COVID-19, however the number of patients in this category is different. They also mention that they evaluated the effectiveness of acetaminophen and NSAIDs, but only evaluated some points related with safety.

→ As your mentioned, we have revised the 1st paragraph in the discussion section. We have removed the sentence, ‘worse outcomes were found in 1,231 patients with COVID-19’. and added the sentence, ‘This study found that 338 of 1,058 patients previously prescribed AAP or NSAIDs had a positive test for SARS-CoV-2.’.

13) Check once again the whole manuscript for any linguistic problems and correct typo mistakes. For example, there are some errors in the spelling of the abbreviation NSAIDs.

→ We have checked once again the whole manuscript for linguistic problems and corrected typo mistakes and the spelling of the NSAIDs.

Reviewer 2 Report

This study compares the effect of paracetamol with NSAIDS on COVID-19 hospitalized patients at early development of the pandemics. This study is well-designed and performed. The main drawback is that the authors do not differentiate between different types of NSAIDS. They overall conclude at safety of both categories of drugs.

Similar conclusions were reached by other authors, see for example   doi: 10.1007/s40265-022-01822-z. and doi: 10.3389/fphar.2022.1063246; doi: 10.1111/bcp.15512

There are reports that some of the NSAIDS drugs have therapeutic properties by reducing viral replication in addition to their anti-inflammatory effects. Two drugs, indomethacin and naproxen seem to be beneficial  see for example  Asadi et al, 2021, Terrier et al 2021, Gomeni et al 2020; Kiani et al, 2021

I would recommend attempting getting data with ibuprofen, indomethacin and naproxen as compared to paracetamol, not only with respect to safety but also with respect to efficacy. These issues should at least been discussed with appropriate citations.  

Author Response

→ This study has some limitations such as the dose regiment of treatment acetaminophen or NSAIDs, so we couldn’t get each different types of NSAIDs from the Korean Health Insurance. Unfortunately, we have analyzed the therapeutic properties depending on each drug, so we have discussed in the limitations of discussion section. And I have demonstrated and cited references as you mentioned in discussion section.

‘And recent systematic review, Zhao et al [18] also demonstrated prior use of NSAIDs was not associated with mechanical ventilation, but with a decrease mortality[aOR], 0.68; 95% confidence interval [CI], 0.52-0.89).’

‘There have been several studies that revealed the efficacy of different types of NSAIDs. In double-blinded randomized control study, 500mg naproxen every 12 hours could improve cough and shortness breath in COVID-19 patients.[20] In vitro study [21], compared to paracetamol or the COX-2 inhibitor celecoxib, naproxen has direct antiviral activity against SARS-CoV-2 replication and protects the lung epithelium from damage caused by the pandemic virus, combining antiviral and anti-inflammatory effects. Another study reported the effectiveness of an in vitro study according to the dose of indomethacin, the treatment with sustained-release formulation at a dose of 75 mg twice daily is expected to achieve a complete response within 3 days for the SARS-CoV-2 infection.[22] And Kiani et al.[23] investigated that the effectiveness of ketotifen, naproxen and indomethacin alone or in combination in reducing SARS CoV-2 replication. They found that the combination of ketotifen with indomethacin or naproxen all increased in percentage of SARSA-CoV-2 replication and no cytotoxic effects were observed. Although this study did not analyze whether different types of NSAIDs affect serious clinical outcomes, it was found that NSAIDs had a positive effect on COVID-19 infection when reviewing previous studies.’

Round 2

Reviewer 1 Report

Dear Editor and Dear authors

The corrections made to the manuscript are adequate, however I recommend attending to a suggestion in the discussion section before considering publication.

 1)The new information that was added in discussion section is contradictory and does not correspond to that cited in the reference 23. Also review the spelling of the abbreviation SARS-CoV-2.

And Kiani et al.[23] investigated that the effectiveness of ketotifen, naproxen and indomethacin alone or in combination in reducing SARS CoV-2 replication. They found that the combination of ketotifen with indomethacin or naproxen all increased in percentage of SARSA-CoV-2 replication and no cytotoxic effects were observed.

Author Response

The corrections made to the manuscript are adequate, however I recommend attending to a suggestion in the discussion section before considering publication.

 1)The new information that was added in discussion section is contradictory and does not correspond to that cited in the reference 23. Also review the spelling of the abbreviation SARS-CoV-2.

And Kiani et al.[23] investigated that the effectiveness of ketotifen, naproxen and indomethacin alone or in combination in reducing SARS CoV-2 replication. They found that the combination of ketotifen with indomethacin or naproxen all increased in percentage of SARSA-CoV-2 replication and no cytotoxic effects were observed.

→Thank you again for your valuable comments. We have corrected the sentence and checked the spelling of the abbreviation SARS-CoV-2.

‘And Kiani et al.[23] investigated that the effectiveness of ketotifen, naproxen and indomethacin alone or in combination in reducing SARS CoV-2 replication. They found that the combination of ketotifen with indomethacin or naproxen all increased in percentage of SARSA-CoV-2 replication and no cytotoxic effects were observed.’ to ‘And Kiani et al.[23] investigated that the effectiveness of ketotifen, naproxen and indomethacin alone or in combination in reducing SARS CoV-2 replication. They found that the combination of ketotifen with indomethacin or naproxen all increased in percentage inhibition of SARS-CoV-2 replication and no cytotoxic effects were observed.’

Reviewer 2 Report

It is a pitty that you cannot get more details  

Ok with your modifications in the Discussion. 

Author Response

It is a pitty that you cannot get more details  

Ok with your modifications in the Discussion.

→ We really appreciated your valuable comments.